# Substrate Specificity and Structural Modeling of Human Carboxypeptidase Z: A Unique Protease with a Frizzled-Like Domain

**DOI:** 10.3390/ijms21228687

**Published:** 2020-11-18

**Authors:** Javier Garcia-Pardo, Sebastian Tanco, Maria C. Garcia-Guerrero, Sayani Dasgupta, Francesc Xavier Avilés, Julia Lorenzo, Lloyd D. Fricker

**Affiliations:** 1Institut de Biotecnologia i Biomedicina and Departament de Bioquimica i Biologia Molecular, Universitat Autònoma de Barcelona, 08193 Bellaterra, Barcelona, Spain; javiergarciapardo@msn.com (J.G.-P.); sebastiantanco@gmail.com (S.T.); mariacarmen.garciag@gmail.com (M.C.G.-G.); francescxavier.aviles@uab.es (F.X.A.); 2BiosenSource BV, B-1800 Vilvoorde, Belgium; 3Department of Molecular Pharmacology, Albert Einstein College of Medicine, Bronx, New York, NY 10461, USA; capri181@gmail.com

**Keywords:** carboxypeptidase Z, metallocarboxypeptidase, substrate specificity, Wnt signaling, frizzled, cysteine rich domain, growth factor

## Abstract

Metallocarboxypeptidase Z (CPZ) is a secreted enzyme that is distinguished from all other members of the M14 metallocarboxypeptidase family by the presence of an N-terminal cysteine-rich Frizzled-like (Fz) domain that binds Wnt proteins. Here, we present a comprehensive analysis of the enzymatic properties and substrate specificity of human CPZ. To investigate the enzymatic properties, we employed dansylated peptide substrates. For substrate specificity profiling, we generated two different large peptide libraries and employed isotopic labeling and quantitative mass spectrometry to study the substrate preference of this enzyme. Our findings revealed that CPZ has a strict requirement for substrates with C-terminal Arg or Lys at the P1′ position. For the P1 position, CPZ was found to display specificity towards substrates with basic, small hydrophobic, or polar uncharged side chains. Deletion of the Fz domain did not affect CPZ activity as a carboxypeptidase. Finally, we modeled the structure of the Fz and catalytic domains of CPZ. Taken together, these studies provide the molecular elucidation of substrate recognition and specificity of the CPZ catalytic domain, as well as important insights into how the Fz domain binds Wnt proteins to modulate their functions.

## 1. Introduction

Metallocarboxypeptidases (MCPs) are a group of zinc-containing exopeptidases that cleave single C-terminal amino acids from proteins and peptides [1]. The M14 family of MCPs consists of three subfamilies in humans, each containing six to nine individual members, and all M14 members share a conserved carboxypeptidase (CP) domain. The M14B subfamily of MCPs, also known as the CPN/E subfamily, is composed in humans by eight members that play important roles in the processing of neuropeptides and growth factors [1,2,3]. In addition to the conserved CP domain, all members of the M14B subfamily contain a conserved transthyretin-like (TTL) domain that is not present in the other subfamilies [4,5,6,7,8,9]. The TTL domain possesses a β-sandwich folding related to transthyretin and is required for the production of a catalytically-active enzyme, possibly by contribution to folding and/or protein binding [5,6,9]. Some members within the M14B subfamily are catalytically inactive since they lack one or more amino acids essential for the catalytic mechanism. The five members that are catalytically active, named E, N, M, D, and Z, all show carboxypeptidase B-like substrate specificity, cleaving C-terminal basic residues [2]. Carboxypeptidase E (CPE) primarily functions in the removal of C-terminal lysine and arginine residues from neuroendocrine peptide precursors in the secretory pathway [10]. Carboxypeptidase N (CPN) circulates in plasma and removes C-terminal arginine residues from bradykinin and anaphylatoxins C3a, C4a, and C5a, altering the affinity of these peptides for their receptors [6]. In a similar manner, carboxypeptidase M (CPM) has been proposed to remove C-terminal basic residues from proteins and peptides in the extracellular space [11,12]. Carboxypeptidase D (CPD) is a membrane protein that resides primarily in the trans-Golgi network, where it functions in the same pathway as furin to process proteins into their mature forms [8,9,10]. For example, CPD is necessary for the formation of the active insulin-like growth factor 1 receptor (IGF1R); deletion of the *CPD* gene inhibited signaling through IGF1R and reduced the three-dimensional growth of a lung adenocarcinoma cell line [13].

CPZ is synthesized as a constitutively active enzyme of about 72 kDa, which is secreted through the regulated pathway to the extracellular medium [14,15]. Part of the secreted CPZ is bound to the extracellular matrix (ECM) due to its heparin-binding properties [16]. Previous studies suggested that the presence of a highly conserved C-terminal stretch of ~35 amino acids contributes to the ECM binding [15]. Little is known about the substrate specificity, function, structure and/or interactors of CPZ [2]. Unlike all other MCPs, CPZ contains an N-terminal Fz domain, which is a cysteine-rich sequence present in a number of protein families, including Frizzled receptors, type XVIII collagen, secreted Frizzled-related proteins (sFRP), receptor protein tyrosine kinases, and LDL-receptor class-A domain [2,17,18]. Typically, Fz domains function as receptors for members of the *Wingless* (WG), or its mammalian homolog, the Wnt family proteins [18]. The activation of Frizzled receptors by WG/Wnt proteins leads to the initiation of intracellular signaling pathways (commonly known as Wnt signaling pathways) that mediate vertebrate and invertebrate development and tissue homeostasis by influencing cell proliferation, differentiation, and migration [19,20]. Dysregulation of Wnt signaling has been associated with a variety of human pathologies such as several hereditary diseases, osteoarthritis, and cancer [21,22].

The physiological role of the Fz domain in CPZ remains a mystery, and it was initially proposed to have a role in Wnt signaling based on the localization of CPZ in the ECM as well as on the dynamic expression of CPZ during development [2]. Several proteins containing Fz domains, such as sFRP, can act as Wnt modulators, inhibiting and/or enhancing its biological activities and influencing gene expression [23,24]. Two studies have investigated the ability of CPZ to bind Wnt-4 and modulate its signaling in skeletal elements in chicken [25,26], demonstrating that CPZ might regulate the terminal differentiation of growth plate chondrocytes in vitro [26]. Wnt-4 and several other human Wnts have C-terminal Arg or Lys residues, and these could potentially be removed by CPZ.

In the work presented here, we have used a combination of fluorescent substrates and quantitative peptidomics approaches to gain insights into the substrate specificity of human CPZ, and to elucidate the contribution of the N-terminal Fz domain to its catalytic activity. Furthermore, we performed structural modeling of the catalytic domain, as well as the Fz domain of human CPZ, to dissect its structural features. From these studies, we found that CPZ is a metallocarboxypeptidase that specifically cleaves substrates and peptides with C-terminal basic amino acids, especially those containing C-terminal Arg residues. Moreover, we have characterized the structure and functions of the Fz domain within CPZ, through the study of the activity of an N-terminal truncated form of CPZ and the study of its structural characteristics. Overall, our work on human CPZ provides an extensive and detailed description of its substrate preferences and suggest a direct binding of the Fz domain of CPZ to Wnt proteins.

## 2. Results

### 2.1. Recombinant Protein Expression and Purification

Although CPZ is a secreted enzyme, it binds with high affinity to the ECM. Due to this characteristic, it was difficult to purify the protease. To solve this problem, we deleted the C-terminal 29 residues and expressed the recombinant enzyme in suspension-growing HEK293F cells that were cultured in medium containing heparin (Figure 1).

Using this production strategy, we were able to produce high levels (>1 mg/L) of two different constructs of human CPZ (Figure 1). The deletion of the highly basic 29-residues long C-terminal tail increased the amount of protein expressed without altering its activity [27]. Accordingly, two forms of human CPZ lacking the C-terminal region, with or without the Fz domain (hereafter termed as CPZ and CPZΔFz, respectively), were cloned into the pTriEx^TM^-7 expression vector to encode an IgM secretion signal sequence and an N-terminal Strep-Tag II (Figure 1). Both CPZ-pTrieX-7 and CPZΔFz-pTrieX-7 constructs were transfected into the HEK293F cells for transient expression. After 48 h, recombinant proteins were detected on the extracellular media, showing major bands of SDS-PAGE with molecular weights of ~70 kDa and ~55 kDa (corresponding to CPZ and CPZΔFz, respectively). Maximum yields of soluble CPZ at the extracellular medium were detected after 10 days, displaying protein levels up to 2–3 mg/L in the extracellular conditioned medium. Both proteins were purified from the extracellular conditioned medium by applying a three-step purification protocol (see Figure 2). The first chromatographic step was a heparin-based affinity chromatography. The eluates containing either CPZ or CPZΔFz from the first purification step were pooled and purified over an anti-Strep tag affinity resin and further fractionated on a size exclusion column (Figure 2). In the size-exclusion chromatography step, the purified proteins elute mainly as single peaks and showed apparent masses in agreement with its monomeric molecular weights in solution [27].

### 2.2. Enzymatic Characterization of CPZ Using Fluorescent Synthetic Substrates

The effect of pH on CPZ and CPZΔFz carboxypeptidase activity was evaluated using the fluorescent substrate dansyl-Phe-Ala-Arg. The optimal pH range of the CPZ protein was between 7.0 and 8.0, with >50% activity over the pH range between 6.0 and 8.5 (Appendix A). The CPZ form without the Fz domain (CPZΔFz) showed similar behavior with the same optimal pH range (Appendix A).

The enzymatic activities of CPZ were tested against three different dansylated tripeptides. To compare substrates, different amounts of the purified enzyme were incubated with 200 µM of each fluorescent substrate in a 100 mM Tris-acetate, pH 7.5, 100 mM NaCl buffer, and the relative amount of product determined as described in the experimental section. Of the substrates examined, only dansyl-Phe-Ala-Arg was efficiently cleaved by CPZ under the experimental conditions. We also compared the activity of CPZ with that of human CPD. While 20% of dansyl-Phe-Ala-Arg was cleaved by CPZ, the latter enzyme was able to cleave three times more substrate under the same conditions (Appendix A). It is worth to mention that no activity was detected towards dansyl-Phe-Gly-Arg or dansyl-Phe-Pro-Arg when assayed with up to 50 nM of CPZ (Appendix A). In contrast, an identical concentration of human CPD cleaved approximately 20% of the dansyl-Phe-Gly-Arg, when tested in similar conditions. The kinetic parameters for CPZ and CPZΔFz were determined using dansyl-Phe-Ala-Arg (Table 1). The *k*cat value obtained for CPZ (recombinant form containing the Fz domain) was 5.3 ± 0.6 s^−1^, which is fairly close to the *k*cat value of CPZΔFz that does not contain the Fz domain (6.2 ± 0.8 s^−1^). CPZ showed a *K*m of 1905 ± 360 μM and a *k*cat/*K*m of 0.0028 ± 0.0008 μM^−1^ s^−1^, both values comparable to the constants determined for CPZΔFz (with *K*m and *k*cat/*K*m values of 1667 ± 385 µM and 0.0039 ± 0.0014 µM^−1^ s^−1^, respectively).

### 2.3. Substrate Specificity Profiling of Human CPZ by Quantitative Peptidomics Appproaches

We applied a combination of two different quantitative peptidomics approaches to study the substrate specificity of human CPZ. In the first experiment, we prepared a peptide library extracted from bortezomib-treated HEK293T cells. The treatment with bortezomib leads to an increase in the total number of peptides found in the cells due to specific proteasome inhibition [29]. As shown in the scheme of the experiment displayed in Figure 3A, different amounts of purified CPZ (i.e., no enzyme, 0.1, 1, 10 or 100 nM) were incubated with the peptide mix extracted from the cells. This peptide library, used to find CPZ substrates, contains hundreds of different endogenous peptides with a wide variety of residues in the C-terminal position. After incubation, the individual reactions were differentially labeled with isotopic variants of 4-trimethylammoniumbutyrate (TMAB) isotopic tags and then combined and analyzed by LC-MS. These tags were synthesized by our group and contained either all hydrogen (D0), 3 deuteriums (D3), 6 deuteriums (D6), 9 deuteriums (D9), or 9 deuteriums and three atoms of 13C (D12) [30,31].

After LC-MS, over 60 peptides were identified through tandem mass spectrometry (MS/MS) and/or by close matching with the theoretical mass, charge, and LC elute time of peptides previously identified by MS/MS.

Detailed analysis of the LC-MS profiles allowed us to identify some peptides which exhibited a peak set with roughly equal peak heights, revealing that these peptides were not substrates or products of CPZ (Figure 3B and Appendix A). Other peptides were extensively cleaved, showing a complete or almost complete decrease in the peak intensity upon incubation with the highest enzyme concentration (i.e., 100 nM) and a partial decrease in the peak intensity with a lower enzyme concentration (i.e., 10 nM); these peptides are considered in our analysis as good substrates of CPZ (Figure 3C and Table 2). Some peptides were only partially cleaved, exhibiting a small decrease in intensity with the highest concentration of enzyme assayed, and no or slight decrease in the peak intensity with the 10 nM enzyme concentration; these peptides are weak substrates of CPZ (Figure 3D and Table 2). A small number of detected peptides showed an increase in the peak intensities correlated with increasing amounts of CPZ; these peptides are considered to be products of CPZ cleavage (Figure 3E and Table 3).

In previous studies, it has been demonstrated that P1′ can be considered as the most important residue to determine the substrate specificity of MCPs, although other residues in positions P1 or even farther from the cleavage site can substantially contribute to the substrate specificity (Figure 4A) [32]. We found that nearly all of the peptides identified as substrates of CPZ contained Arg at the C-terminal (P1′) position, and only one peptide contained a Lys in this position (Figure 4B). Furthermore, all of the peptides identified as substrates for this enzyme contained residues with hydrophobic, small side chains or residues with polar uncharged side chains, such as Ala, Ser, Lys or Leu, in penultimate (P1) position (Table 2). We also identified a large number of non-substrates for CPZ containing a wide variety of C-terminal (P1′) residues such as Leu, Ile, Val, Ala, Phe, Asp, Glu, Gln or Asn (Appendix A). Four peptides were identified as products of CPZ; these peptides resulted from cleavage of Lys or Arg from the C-terminus and with Ala, Phe, Val, or Leu at the P1 position of the cleavage site (Table 3). Taken together, these data suggest that that CPZ only cleaves substrates with either C-terminal Arg or Lys amino acids.

The HEK293T peptide library used in the first peptidomics analysis contained a broad range of C-terminal residues, which was useful to evaluate the specificity for the P1′ position. Once that was established, we performed a second experiment in which different amounts of purified CPZ were incubated with a tryptic peptide library generated upon the digestion of five selected purified proteins (see experimental scheme in Figure 5A). A similar tryptic peptide library was used before to characterize the substrate specificities of the first and second domains of human CPD [8]. This library contains mainly peptides with basic C-terminal residues and has a variety of amino acids in the P1 position. In this second experiment, we identified >50 peptides through tandem mass spectrometry (MS/MS) and/or close matches with the mass and charge of predicted tryptic peptides.

Consistent with the previous experiment, some of these peptides from the tryptic library were not substrates or products of CPZ, despite the presence of C-terminal Lys or Arg residues on nearly every peptide in the library (Figure 5B and Appendix A). By contrast, one peptide was extensively cleaved, showing a complete or almost complete decrease in the peak intensity upon incubation with the highest concentration of enzyme (100 nM) and a partial decrease in the peak intensity with a lower concentration (i.e., 10 nM); this is considered as a good substrate of CPZ (Figure 5C and Table 4). Five peptides were only partially cleaved, exhibiting a small decrease in intensity with the highest concentration of enzyme assayed, and no or slight decrease in the peak intensity with the concentration of 10 nM CPZ; these are considered as weak substrates of CPZ (Figure 5D and Table 4). Two peptides showed an increase in peak intensities that correlated with the amount of CPZ; these are considered as products of CPZ cleavage (Figure 5E and Table 4).

Analysis of the C-terminal (P1′) residue of CPZ substrates versus non-substrates, showed again a strict preference for Lys and Arg amino acids (see Figure 6A). Nevertheless, a large number of peptides with C-terminal Arg and Lys were identified as non-substrates, probably as a result of the influence of the residue in the P1 or other positions (Appendix A). Therefore, we analyzed the influence of the penultimate (P1) residue on CPZ substrates and non-substrates (see Figure 6B). The best peptide substrate identified for CPZ contains a Lys amino acid in the P1 position, and weak substrates contained C-terminal Ser, Leu, Gly or Thr amino acids (Table 4). No peptides identified as CPZ substrates contained Pro, Asp, Gln, Phe, Val, Ala, Asn, Glu, or Ile amino acids at P1 position (see Table 4). We also identified two peptides as products for CPZ. These peptides were derived from the proteolytic cleavage of tryptic peptide precursors with C-terminal Arg residues and with Phe or Lys residues in the P1 position (Figure 5E and Table 4).

Given the specificity of CPZ against C-terminal basic residues, we further assessed whether CPZ has a preference for the cleavage of Lys versus Arg. We examined the activity of the enzyme against synthetic Met-enkephalin-derived heptapeptides with a Lys or Arg in the P1′ position and Lys at P1 (YGGFMK**R** or YGGFMK**K**, respectively). Both peptides were incubated in two independent reactions with the purified CPZ and the samples were monitored by MALDI-TOF mass spectrometry after different time points of incubation. While the enzyme completely cleaved off the Arg residue after 300 min of incubation (Figure 7A), only a small proportion of the Met-enkephalin peptide containing a C-terminal Lys was processed (Figure 7B). We also investigated whether the presence of Lys or Arg amino acids in the P1 position can influence the activity of CPZ. For this purpose, we incubated the purified CPZ with a Met-enkephalin peptide containing Arg instead of Lys in this position (YGGFM**R**R). CPZ cleaved more efficiently the peptide with a Lys in P1 (Figure 7C). As expected, the enzyme was not able to process a peptide with a C-terminal acidic residue (i.e., Glu) under equivalent experimental conditions (Figure 7D).

### 2.4. Structural Modeling of the Catalytic and TTL Domains of Human CPZ

To gain more insights into the structure and function relationships of human CPZ, we modeled CPZ without its N-terminal Fz domain, based on previously solved crystal structures of human CPN (PDB 2NSM), human CPM (PDB 1UWY), *Drosophila melanogaster* CPD domain I (PDB 3MN8) and duck CPD domain II (PDB 1H8L) by using I-TASSER [33]. The resultant model is shown in Figure 8A, which displays both the sequential and topological similarity with the rest of M14B MCPs available structures. Among them, CPN has the highest sequential identity (51%), as well as topological similarity (with a root mean square deviation of 0.48 Å) to human CPZ. In the CPZ model, two independent domains are clearly visible, the M14 CP central domain and the typical C-terminal transthyretin-like (TTL) domain found in all members of the M14B subfamily of MCPs, which shares topological similarity and connectivity with transthyretin (Figure 8A).

The predicted pI of human CPZ (including only the carboxypeptidase and the transthyretin-like domain) is ~8.3. This pI value rises to ~9.3 when only the CP domain is considered. This high pI may be responsible for the ECM location and heparin-binding properties of CPZ. As shown in Figure 8B, the representation of the electrostatic potential into the CPZ surface shows that the majority of basic residues are located on the face of the enzyme opposite to that of its active site, suggesting that this surface region may orient CPZ with respect to the ECM and facilitate access of substrates to the catalytic cleft.

A detailed analysis of the catalytic site of human CPZ (Figure 8C,D) shows that the residues involved in the binding of the catalytic zinc ion His69, Glu72, and His196 (according to bCPA numeration) are conserved. Similarly, those residues directly involved in catalysis (i.e., Arg145 and Glu270 in bCPA) are also conserved. Nonetheless, some residues involved in substrate binding and specificity (i.e., Leu203, Gly207, and Ile255 in bCPA) are substituted in CPZ by Ser387, Asp391, and Ser457, respectively (Figure 8C,D). These residue substitutions are also found in other MCPs like the first domain of human CPD, which contains Ser and Asp residues in positions homologous to Leu203 and Gly207 of bCPA, whereas the second domain of CPD contains Asn in an equivalent position to Leu203. As a noteworthy difference, CPZ has a Ser residue in a position equivalent to Ile255 in bCPA, while both CPD domains have a Gln amino acid in this position (Figure 8C,D). These are the sites that are expected to govern substrate specificity in CPZ.

### 2.5. Structural Modeling of the CPZ Fz Domain: Insights into Its Structure and Wnt Recognition

To date, a reduced number of crystal structures of Fz domains have been solved [34,35,36,37]. Because the three-dimensional structure of the Fz domain of human CPZ is still unknown, here we modeled it based on previously solved structures from the cysteine-rich Fz domains of mouse Fz8 (PDB 4F0A) and mouse sFRP3 (PDB 1IJX). The derived three-dimensional model is shown in Figure 9A.

The Fz domains of mouse Fz8 and sFRP3 share topological similarity and connectivity with the Fz domain of human CPZ. The root mean square deviation calculation between the cysteine-rich domain of mouse Fz8 and our modeled structure gave a value of 1.66 Å, despite sharing only 28% of sequence identity (Figure 9B,C). The CPZ Fz domain is mainly composed of four α-helices stabilized by ten conserved cysteines forming five disulfide bonds. In addition, two short β-strands at its N-terminus form a minimal β-sheet with β2 passing through a knot created by disulfide bonds (Figure 9A). No major differences were observed between the three-dimensional structures of the cysteine-rich Fz domain of mouse Fz8 and the Fz domain of human CPZ. Slight differences are mainly observed in the length and limits of the first and second α-helix (Figure 9B,C).

Due to the structural similarity between the Fz domain of human CPZ and the extracellular domain of mouse Fz8, we used the solved structure of the complex between mouse Fz8-CRD and Xenopus Wnt-8 (XWnt-8) to model the interaction of our modeled structure with Wnt-8 (Figure 8D). To gain insights into Wnt recognition by the CPZ Fz domain, we analyzed the most important structural elements of this domain for Wnt binding. In the resultant complex, XWnt-8 appears to grasp the Fz domain of CPZ at two opposing sites using an extended “lipid-modified thumb” and “index finger” projecting from a central “palm” domain. The two contact sites are well defined and named “Site 1” and “Site 2 (see Figure 9D). A detailed analysis of Site 1 shows that the surface clef of Fz8 that accommodates the 16-C palmitoleic acid extended from the XWnt-8 Ser197 (the “lipid-modified thumb”) is conserved in the Fz domain of human CPZ. This surface cleft is shaped by helix 1, 2, and 4, as well as the first loop immediately after helix 4 of the Fz domain. In CPZ, this surface cleft is lined by hydrophobic amino acids, as previously described for Fz8-CRD and other Fz domains [35]. Site 2, located at the opposite side of Site 1 in the CPZ Fz domain, is also conserved. This second interaction site is formed by a depression between inter-helical loops on the Fz domain and is responsible for the accommodation of the XWnt-8 finger loop (the “index finger”).

The C-terminal tail of the Fz domain is connected with the N-terminal tail of the CPZ CP domain through a ~10 amino acid segment. In the Fz-CPZ/XWnt-8 complex, the C-terminal tail of the Fz domain is extended to the side of the main plain of the structure, and oriented to the C-terminal tail of XWnt-8. Although the structural arrangement between the Fz domain and the catalytic domain of CPZ is still unknown, this finding indicates the possibility that the catalytic domain remains close to the C-terminal residue of Wnt proteins, thus facilitating the cleavage of its C-terminal basic amino acids after Wnt recognition (Figure 9D).

## 3. Discussion

A major focus of the present study was the detailed screening of the cleavage specificity of human CPZ, which has not been examined so far, and the elucidation of its molecular basis through modeling. A previous study showed that this enzyme has a strong preference for substrates with C-terminal Arg, as deduced from a single assay using a pair of synthetic substrates [15]. In the present study, we used a series of defined synthetic substrates as well as quantitative peptidomics approaches with a broad spectrum of peptides to characterize the substrate specificity of the enzyme. One previous major limitation was the difficulty in obtaining enough purified CPZ for analysis. We solved this limitation through the use of a recombinant approach in the presence of heparin, which allowed us to produce milligrams of pure and active enzyme.

The synthetic substrate dansyl-Phe-Ala-Arg enabled us to determine that human CPZ has maximum activity at neutral pH (i.e., pH 7.5). This observation is in agreement with previous studies on human CPZ, which tested the activity of this enzyme against a couple of dansylated tripeptides differing in the P3 position [15]. Further, the determination of catalytic constants revealed that CPZ is an enzyme with low intrinsic activity. Using dansyl-Phe-Ala-Arg, CPZ has a *kcat*/*Km* value of 0.0028 ± 0.0008 µM^−1^ s^−1^ and a *Km* of 1905 ± 360 µM. When these *kcat*/*Km* values are compared with those of other enzymes of the same subfamily, human CPD appears to be ~30-fold more efficient than human CPZ in the cleavage of the same substrate under similar experimental conditions [8]. A comparison with each individual isolated CPD domain reveals that while the first domain of human CPD shows ~10-fold more activity than CPZ, the second domain is only ~3-fold more active. Similarly, the well-known CPE also appear to be 136-fold more efficient than human CPZ in the cleavage of this substrate [28].

Synthetic substrates are useful to investigate the substrate–product relationships and to determine the kinetic parameters but are limited by the expense and the time required for the measurement of each substrate. In the present study, we tested a large number of peptides using a combination of TMBA-isotopic labelling and mass spectrometry quantitative peptidomics approaches, allowing us to perform in-depth characterization of the substrate specificity of CPZ. These analyses showed that CPZ is specific for cleaving Arg or Lys from the C-terminus, with no other residue permissible in the P1′ position. When nearly identical peptides were compared, differing only in the C-terminal residue, CPZ demonstrated a preference for Arg over Lys. In the P1 position, CPZ showed a preference for Lys, Gly, Ser, Thr, and Leu. A limitation of the peptidomics study is that Cys was not present in any of the detected peptides; this is a relatively rare amino acid, and it was previously noted that Cys is even further under-represented in peptidomics databases [31]. CPZ has previously been shown to cleave the C-terminal Arg residue from Wnt-4, and the penultimate residue is Cys [25,26], and therefore this residue should be added to the list of permissible P1 substrates of CPZ. From the present study and the finding that Lys is permissible in the P1′ position, this raises the possibility that CPZ cleaves additional Wnt proteins that contain Lys on the C-terminus.

From the evolutionary point of view, it appears that CPZ only required a moderate drift in active site residues common in the M14 family and M14B subfamily to generate its differential specificity. Thus, besides an absolute conservation of catalytic and Zn binding sites, this enzyme displays a few moderate but very selective changes in the substrate specificity pocket, such as Leu203 and Ile255 in bovine CPA (bCPA) that are replaced in CPZ by Ser387 and Ser457, respectively. Similarly, the human CPD domain I contains Ser264 in a position equivalent to Leu203 of bovine CPA1 and Ser198 of CPM, whereas CPD domain II contains Asn678 in an equivalent position. The role of these residues in determining the substrate specificity of CPD has been previously studied in detail by performing docking experiments with model peptide substrates [8]. The presence of Ser264 in the domain I of CPD was associated with a higher preference of this domain for peptide substrates with C-terminal Arg [8]. A different question is whether such amino acid substitutions contribute to the low activity of CPZ against the standard substrate (dansyl-Phe-Ala-Arg), in comparison with the rest of the MCPs that cleave C-terminal Arg residues. It is possible that residue variations in the loops surrounding the border of the catalytic funnel and S1 subsite in CPZ may reduce the access of natural substrates to the catalytic cleft.

CPZ is unique among MCPs due to the presence of an N-terminal cysteine-rich domain of ~120 amino acids with a ~30% sequential homology to the Fz domain within the Frizzled protein family of Wnt receptors [38]. Only a small number of protein families contain Fz domains and many of these proteins have been shown to bind Wnt proteins [18]. This is the case of the Frizzled family of seven-pass transmembrane receptors that, following Wnt binding, activate intracellular signaling events [18]. Another group of related proteins with cysteine-rich domains are the secreted Frizzled Related Proteins (sFRP). This family of secreted molecules act typically as Wnt inhibitors [24]. From our structural model of the Fz domain, we found that the Fz domain of human CPZ has structural similarity and connectivity with the cysteine-rich domain of Frizzled receptors (taking as an example the mouse Fz8-CRD) and conserves the most important elements for Wnt recognition. This issue reinforces the idea that some M14 carboxypeptidases, including CPZ, can be involved in multiple functions independent from its catalytic activities. For example, CPE was proposed to be a positive/negative modulator of the Wnt signaling pathway through its binding to the Wnt-3A-Frizzled receptor complex [39,40]. Accordingly, the Fz domain of CPZ might also have independent functions without the requirement of its catalytic activity. In agreement, we found that the removal of the N-terminal Fz domain in CPZ does not greatly affect its enzymatic activity as a carboxypeptidase against dansyl-Phe-Ala-Arg, showing the CPZΔFz variant has comparable kinetic parameters to the wild-type enzyme which contains the Fz domain.

The presence of the Fz domain in CPZ, together with its dynamic expression pattern during development and its localization in the ECM, may suggest a special affinity for Wnt molecules. It is intriguing that the majority of human Wnt proteins are predicted to contain C-terminal basic residues, immediately after the last conserved Cys amino acid, out of the globular cysteine-rich core (Figure 10). It is unclear whether the major forms of these proteins that exist in vivo contain the C-terminal basic residue, or whether it has been removed by proteases. As discussed above, in the present study, we found that CPZ can remove C-terminal basic residues from a wide variety of peptides, implying that these molecules are potential substrates for CPZ. If processing does occur at this site, it is not known whether this affects the biological activity. In a more general role, the C-termini of proteins represents a unique region that contains mini-motifs, short peptides with an encoded function generally characterized as binding, posttranslational modifications, or trafficking. Many of these activities can be regulated by C-terminal proteolysis [41,42,43,44]. As an example, in the case of human Epidermal Growth Factor (EGF), the cleavage of three or more C-terminal residues lead to the loss of its biological activity, since these C-terminal residues are important for receptor binding [45,46,47].

Noteworthy, the C-terminal tail of Wnt proteins does not contact Fzd8-CRD in the XWnt-8-Fzd8 modeled structure. Nonetheless, it is interesting to speculate that the C–terminal residue/s in Wnt proteins might serve for recognition and/or stabilization of the interactions with co-receptors (e.g., LRP binding) or alternatively serve for other structural/functional purposes. Thus, it is possible that removal of the C-terminal residue activates or inactivates the Wnt, renders it susceptible to further degradation, or alters the targeting of the Wnt within the extracellular matrix. In support of this, the polarized nature of our model for CPZ suggests that this enzyme may orient itself, facing back with respect to the negatively charged glycoproteins of the ECM, remaining its active site accessible for potential substrates. It is interesting that a similar polarity has also been found for other carboxypeptidases with similar heparin-binding properties, such as mast cell carboxypeptidase or human carboxypeptidase A6 [48,49].

Taken together, our results demonstrate that CPZ is able to cleave with high efficiency a subset of peptides substrates with Arg or Lys on the C-terminus, and does not cleave peptides with non-basic C-terminal residues. Furthermore, CPZ has a moderate preference for substrates containing basic, hydrophobic, or polar uncharged side chains in the P1 position. These findings, together with the structural analysis of the Fz domain and the catalytic domain of CPZ, revealed that this enzyme could play a role in the binding or processing of Wnt signaling molecules, acting as an effector. Nonetheless, further functional studies are needed to test these hypotheses and unravel the exact biological functions of this intriguing enzyme in the Wnt signaling pathway.

## 4. Materials and Methods

### 4.1. Cell Culture

HEK293T cells (ATCC CRL-3216) were maintained in Dulbecco’s Modified Eagle’s Medium (DMEM) further supplemented with GlutaMAX and with 10% (*v/v*) fetal bovine serum (Invitrogen, Thermo Fisher Scientific, Waltham, MA, USA) at 37 °C, 10% CO2, and 95% humidity. HEK293F cells (FreeStyle 293F cell line, Thermo Fisher Scientific, Waltham, MA, USA) were grown in FreeStyle 293 expression medium (Invitrogen, Thermo Fisher Scientific, Waltham, MA, USA) in cell culture flasks on a rotary shaker at 120 rpm, 37 °C, 8% CO2, and 70% humidity.

### 4.2. Recombinant Protein Production and Purification

The cDNA encoding human CPZ was amplified by PCR using the full-length human CPZ cDNA as template. Two different fragments were cloned into the pTriExTM-7 vector for its heterologous expression in mammalian cells. The first form, containing the catalytic CP domain, the TTL domain, and the Fz domain (residues 19-623), named here as CPZ, was cloned into the pTriExTM-7 expression vector (Merck Millipore, Burlington, MA, USA) to encode a mouse IgM secretion signal sequence and an N-terminal Strep-Tag^®^II fusion protein. Similarly, the second CPZ form without the N-terminal Fz domain (residues 186-623) and the sequence of human CPD (residues 32–1298) were cloned into the same expression vector and named as CPZΔFz and CPD, respectively. The expression of CPZ and CPZΔFz was carried out in mammalian cells, in a resembling manner as recently described for other MCPs [8,46]. In brief, DNA transient transfections of CPZ and CPZΔFz were achieved using 25 kDa polyethylenimine (PEI, Polysciences, Warrington, PA, USA), in a ratio of 1:3 (µg DNA/µg PEI). HEK293F cells were diluted to a cell density of about 0.5 × 10^6^ cells/mL, grown for 24 h and then transfected with 1 µg of DNA per ml of culture. After 48 h transfection, sodium heparin was added to the culture, and then cells incubated for additional 6–8 days. Immediately before protein purification, the culture supernatant was centrifuged, filtered through 0.22 µm filters and bound to a heparin-affinity chromatography resin (Heparin HyperD^®^, PALL Life Sciences, Port Washington, NY, USA). Protein elution was carried out with an increasing gradient of NaCl (from 0 to 1 M) in a 100 mM Tris-HCl, pH 8.0 buffer. The eluted fractions containing CPZ were detected by SDS-PAGE, pooled and loaded onto a Strep-tag affinity column (IBA-Lifesciences, Gottingen, Germany) previously equilibrated with binding buffer (100 mM Tris-HCL, pH 8.0, 150 mM NaCl buffer), washed with 5 column volumes of binding buffer and eluted with the same buffer containing 2.5 mM d-desthiobiotin (IBA-Lifesciences Gottingen, Germany). All the eluted fractions were analyzed by SDS-PAGE and the samples with the purest enzyme were pooled and loaded onto a size exclusion chromatography (HiLoad Superdex 75 26/60 column, GE Healthcare, Chicago, IL, USA) previously equilibrated with a 50 mM Tris-HCL, pH 7.5, 150 mM NaCl buffer. Expression and purification of CPD were carried out as described elsewhere [8]. The final purified proteins were concentrated and flash-frozen at approximately 0.3 mg/mL and stored at −80 °C.

### 4.3. HEK293T Bortezomib Treatment and Peptide Extraction

Peptides from HEK293T were obtained as described previously [8]. Briefly, HEK293T cells were grown to 70% confluence in 150 mm cell culture plates in a DMEM medium supplemented with 10% FBS and containing a mixture of penicillin and streptomycin antibiotics. HEK293T cell plates were treated with fresh media containing 0.5 µM bortezomib for 1 h at 37 °C. For peptide extraction, cells were washed three times with cold Dulbecco’s phosphate-buffered saline (DPBS, Invitrogen, Thermo Fisher Scientific, Waltham, MA, USA), immediately scraped and centrifuged at 8000× *g* for 5 min. The cell pellet was resuspended in 1 mL of 80 °C water and the mixture was incubated for 20 min in an 80 °C water bath. Then, samples were cooled, transferred to 2 mL low retention microfuge tubes and centrifuged at 13,000× *g* for 20 min. The soluble fractions containing HEK293T cell peptides were stored at −80 °C overnight. These samples were thawed, centrifuged again as above, and the supernatants were collected and concentrated in a vacuum centrifuge to a final volume of 1.5 mL. Finally, samples were cooled, acidified with HCl, centrifuged at 13,000× *g* for 40 min at 4 °C and the supernatants stored at −80 °C until reactions were performed.

### 4.4. Generation of the Tryptic Peptide Library

A tryptic peptide library was generated by digesting five different proteins with trypsin, similarly as described before by our lab [8]. Bovine serum albumin, bovine thryroglobulin, bovine α-lactalbumin, human α-hemogloblin, and human β-hemoglobin were obtained from Sigma-Aldrich and digested with trypsin in separate reactions. The efficiency of protein digestions was determined by SDS-PAGE analysis of the remaining protein. After digestion reactions were stopped by adding 0.1 M HCl, the independent protein reactions were combined and centrifuged at 13,000× *g* for 45 min at 4 °C. The final peptide mixture contained a peptide concentration of about 500 µM, estimated from the starting amount of protein used in the tryptic digestion. The final peptide library was filtered through a 10 kDa centrifugal filter device (Merck Millipore), aliquoted and stored at −80 °C.

### 4.5. Kinetic Measurements Using Fluorescent Synthetic Substrates

Carboxypeptidase activity was evaluated with the fluorescent substrate dansyl-Phe-Ala-Arg, synthesized as described [49]. To perform the experiments, reactions of 100 µL containing 0.2 mM of dansyl-Phe-Ala-Arg and different enzyme concentrations in 100 mM Tris-acetate, pH 7.5, 100 mM NaCl buffer were prepared. All reactions were incubated for 60 min at 37 °C. After incubation, reactions were stopped by adding 50 µL of 0.5 M HCl. Then, 1 mL of chloroform was added to each reaction, and tubes were mixed gently and centrifuged for 2 min at 300× *g*. After centrifugation, 0.5 mL of the chloroform phase from each reaction was transferred to new tubes and completely dried overnight at 25 °C. Finally, dried samples containing the product generated in the enzymatic reaction were solubilized in 200 µL of PBS containing 0.1% of Triton X-100. The amount of product generated was determined by measuring the fluorescence of samples at 395 nm upon excitation at 350 nm using a 96-well plate spectrofluorometer. The catalytic activity of the enzymes was also evaluated against other dansylated tripeptides, such as dansyl-Phe-Gly-Arg and dansyl-Phe-Pro-Arg, synthesized as described [49]. For this, reactions with different amounts of purified CPZ, CPZΔFz, or CPD were incubated as described above for dansyl-Phe-Ala-Arg. For kinetic analysis, purified enzymes were incubated with different substrate concentrations (typically 0, 66, 125, 250, 500, 1000, 1500, and 2500 µM). The amount of enzyme used in the reactions was that ensuring a maximum 20% hydrolysis of the substrate. Kinetic parameters were determined by fitting the obtained data for each enzyme to the Michaelis-Menten equation (Y = (Vmax × X)/(*K*m + X)) by using GraphPad Prism software (v6.03, GraphPad Sofware Inc.: San Diego, CA, USA, 2012) [50], where X is the substrate concentration, Y the enzyme velocity, Vmax the maximum enzyme velocity and *K*m the Michaelis-Menten constant. The pH optimum of CPZ and CPZΔFz was determined with 0.2 mM dansyl-Phe-Ala-Arg in 100 mM Tris-acetate, 150 mM NaCl buffer at the indicated pH values.

### 4.6. Quantitative Peptidomics Analyses

Quantitative peptidomics analyses were performed as described previously for other MCPs [8,51,52]. In essence, peptides obtained either from HEK293T cells or from the tryptic peptide library were incubated for 16 h at 37 °C in 100 mM borate, pH 7.5, 100 mM NaCl buffer with the indicated concentrations of purified CPZ. After incubation, reactions were labeled using standard labeling procedures with 4-trimethylammoniumbutyrate (TMAB) isotopic tags [30,31] activated with N-hydroxysuccinimide (NHS). The tags used in the experiments contained either all hydrogen (D0), 3 deuteriums (D3), 6 deuteriums (D6), 9 deuteriums (D9), or 9 deuteriums and three atoms of 13C (D12). Each sample was labeled with 5 mg of each label dissolved in DMSO over 8 rounds of reaction, by adding 1.6 μL of the label solution to the extract every 20 min. During the first five rounds of labeling, the initial pH of the samples was adjusted to 9.5 with 1 M NaOH. After labeling, unreacted labels were quenched by mixing the samples with 2.5 M glycine. All the individual reactions from a single experiment were pooled, filtered through a 10 kDa centrifugal filter device. TMAB-labeled tyrosines were hydrolyzed by adding 30 μL of 2 M hydroxylamine solution. Finally, samples were desalted through C-18 spin columns (Thermo Fisher Scientific, Waltham, MA, USA) and peptides were eluted with 0.5% TFA and 70% acetonitrile, freeze-dried in a vacuum centrifuge, and subjected to LC-MS as previously described [8]. Data inspection and peptide identifications were performed using Mascot software (Matrix Science, Boston, MA, USA). The MS spectra were manually examined for peak sets reflecting peptides containing various isotopic forms of the TMAB tags. Peptide identifications were considered only if 80% or more of the major fragments from the MS/MS matched predicted b- or y-series fragments. The maximum tolerance accepted to consider the matches was a coincidence with the parent mass within 50 ppm of the theoretical mass, together with an expected charge equal to basic residues plus the N-terminus and a correct number of isotopic tags in the peptide. All the peptidomics analyses were performed at least in duplicate.

### 4.7. MALDI-TOF Mass Spectrometry Experiments

The substrate specificity of CPZ was evaluated against different synthetic peptides. For this purpose, two synthetic Met-enkephalin peptides differing only in the C-terminal residue (YGGFMKR and YGGFMKK) and a Met-Enkephalin with two C-terminal Arg (YGGFMRR) were purchased from Phoenix pharmaceuticals (Phoenix Pharmaceuticals Inc., Mannheim, Baden-Württemberg, Germany). In addition, a fourth synthetic peptide with a C-terminal Glu (ARLSQKFPKAE) was acquired from GenScript (GenScript Biotech, Piscataway, NJ, USA). Approximately 1.6 μM of each peptide was incubated with 100 nM of CPZ in 100 mM Tris-HCl, pH 7.5, 150 mM NaCl buffer at 37 °C up to 24 h. Aliquots of 2 µL were taken after different incubation times (0, 30, 60, 120, and 300 min), immediately stopped by addition of four volumes of 0.1% TFA. Samples from each reaction were mixed with an equal volume of α-cyano-4-hydroxycinnamic acid (hcca) matrix solution, spotted onto a 384 target plate polished steel (Bruker Daltonics, Billerica, MA, USA), and evaporated to dryness at room temperature. Mass spectra were recorded on an ultrafleXtreme MALDI-TOF mass spectrometer (Bruker Daltonics, Billerica, MA, USA) in reflectron positive ion mode, and at 25 kV. A standard peptide calibration mixture (Bruker Daltonics, Billerica, MA, USA) was used as a reference.

### 4.8. Sequence Alignment and Structural Modeling

A structural alignment between the Fz domain of CPZ and the Fz domain of Fz8-CRD was generated using a flexible structure alignment by chaining aligned fragment pairs with twist (FATCAT) algorithm [53] using the protein comparison tool of the RCSB Protein Data Bank. The multiple sequence alignment of the C-terminal tail of human Wnt proteins was carried out using Clustal Omega from the EMBL-EBI [54]. Structural models of the catalytic domain of CPZ and the CPZ Fz domain were constructed by using the automated I-TASSER on-line server [33]. Secondary structure limits were adjusted based on the predictions of Jpred 4 [55]. The best models were selected based on their C-score, which takes into account the significance of threading template alignments and the convergence parameters. These models were further validated using ProSA and PROCHECK [56,57]. PyMOL (v1.3, DeLano Scientific LLC, San Carlos, CA, USA, 2002) [58] was used for the generation of figures and visual inspection of models.

## Figures and Tables

**Figure 1 ijms-21-08687-f001:**
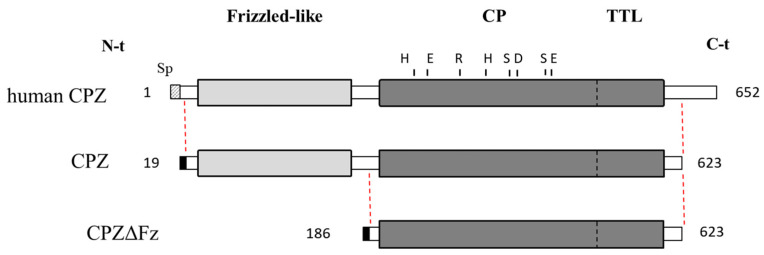
Schematic representation of human metallocarboxypeptidase Z (CPZ) and its recombinant forms. Full-length human CPZ (UniProtKB accession number Q66K79) and the recombinant CPZ forms used in this work are represented. Key residues essential for the catalytic mechanism are indicated: His69, Glu72, Arg145, His198 and Glu270 (numbering based on the mature form of bovine CPA, by convention in the field). Positions of residues Ser387, D391 and Ser457 of CPZ (equivalent to Leu203, Gly207 and Ile255 in bovine CPA, respectively) are also indicated. The recombinant form CPZ lacks 29 residues in its C-terminal region. The recombinant form CPZΔFz, in addition to the C-terminal truncation, lacks 167 residues in its N-terminal tail, corresponding to the Frizzled-like (Fz) domain.

**Figure 2 ijms-21-08687-f002:**
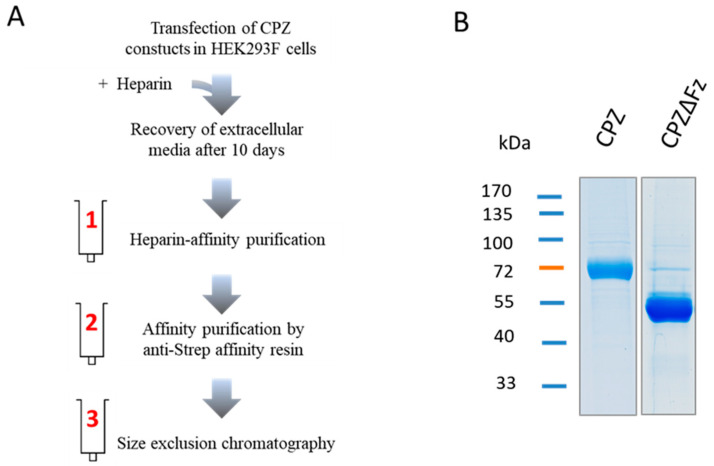
Recombinant expression and purification of human CPZ. (**A**) Schematic diagram of the experimental strategy used for the recombinant expression and purification of CPZ and CPZΔFz. Protein expression was performed by high-level transient transfection in suspension-growing HEK (Human Embryonic Kidney) 293F cells, followed by the addition of heparin at 48 h post-transfection. For protein purification, the extracellular medium was collected after 10-days incubation, and recombinant proteins were purified in three purification steps by heparin-affinity chromatography, affinity chromatography using anti-strep tag affinity resin, and by size-exclusion chromatography; (**B**) SDS–PAGE showing the purity and size of the recombinant proteins.

**Figure 3 ijms-21-08687-f003:**
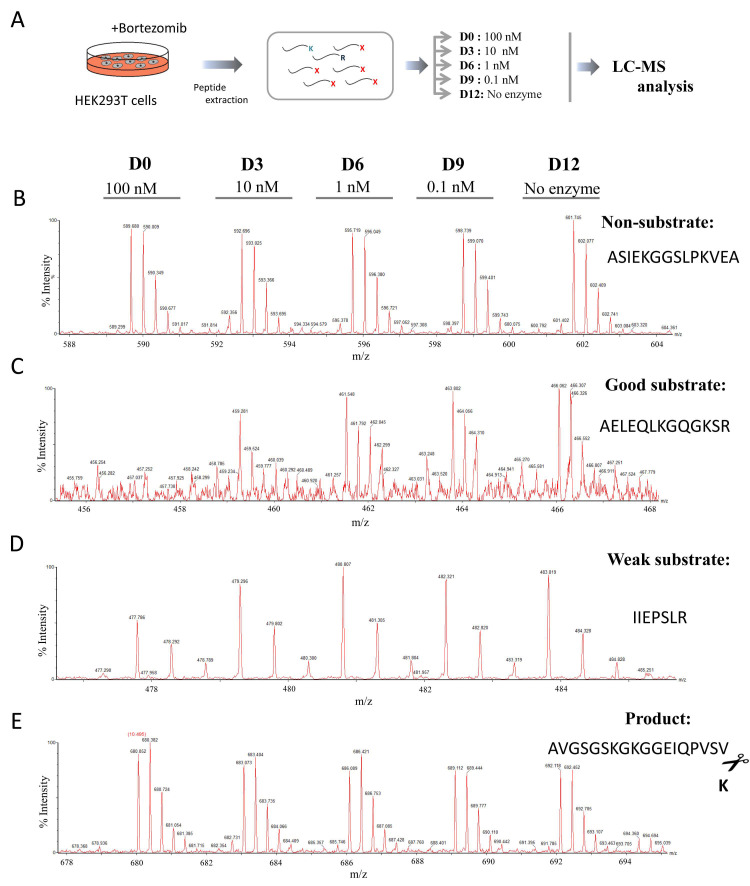
Quantitative peptidomics analysis of CPZ substrate specificity using peptides extracted from HEK293T cells treated with bortezomib. (**A**) Quantitative peptidomics analysis scheme and (**B–E**) representative results. After incubation of the peptides with CPZ at different concentrations, the individual samples were labeled with one of five stable isotopic TMAB tags (D0 for 100 nM CPZ; D3 for 10 nM CPZ; D6 for 1 nM CPZ; D9 for 0.1 nM CPZ; D12 for samples without enzyme). All five reactions were pooled and analyzed by LC-MS. Representative examples are shown for (**B**) non-substrates, (**C**) good substrates, (**D**) weak substrates and (**E**), products of CPZ.

**Figure 4 ijms-21-08687-f004:**
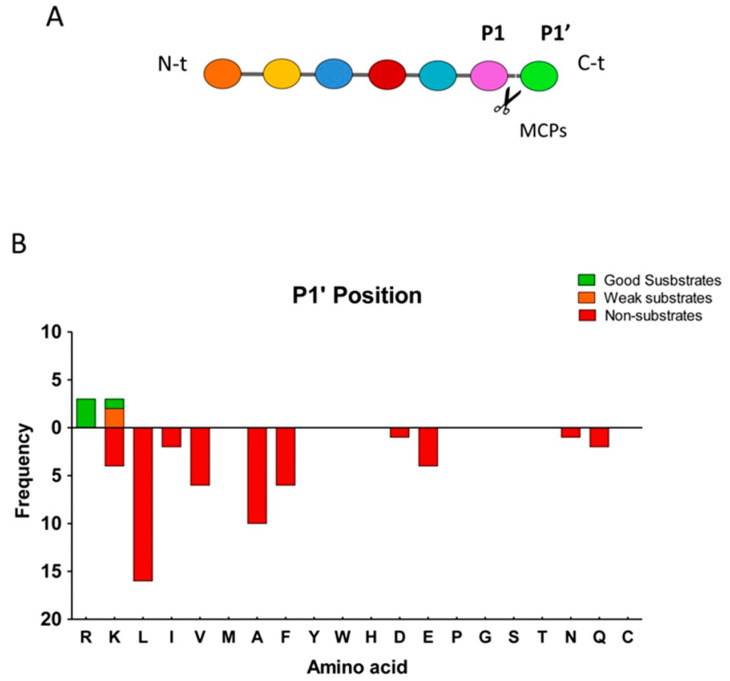
Analysis of the substrate preferences of CPZ from the peptidomics experiment using the HEK293T-derived peptide library. (**A**) Schematic representation of relevant residues involved in a typical MCPs cleavage, according to the model proposed by Schechter and Berger [32]; (**B**) Substrate preference of CPZ at the C-terminal (P1′) position. The frequency for each amino acid found in P1′ is indicated for good substrates, weak substrates, and non-substrates.

**Figure 5 ijms-21-08687-f005:**
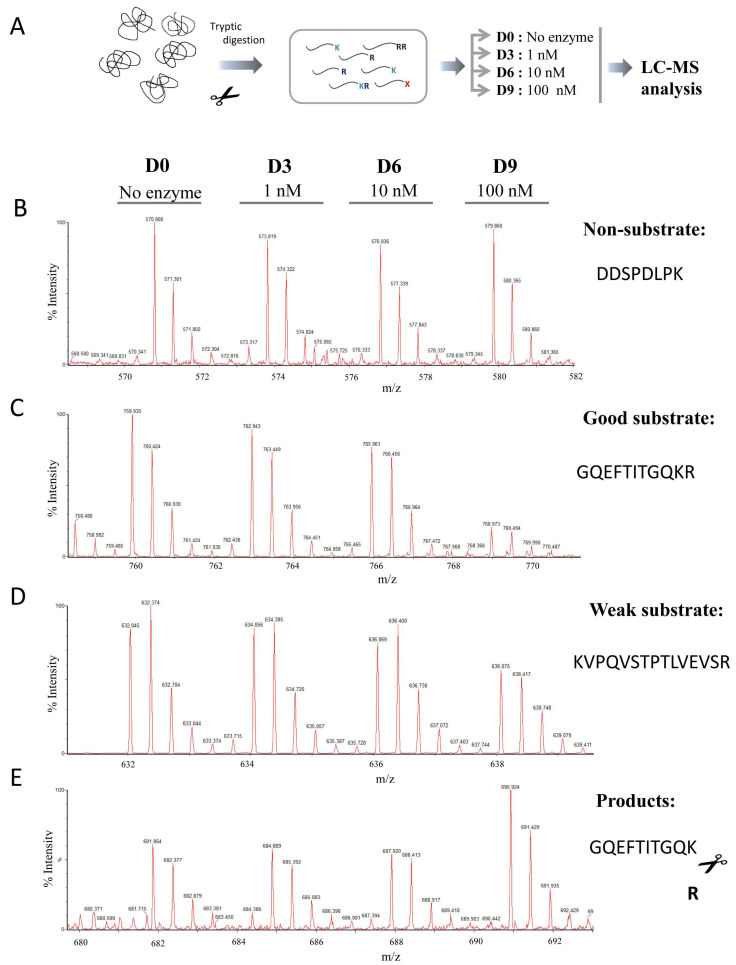
Quantitative peptidomics analysis scheme for the characterization of the substrate specificity of CPZ using a tryptic peptide library. (**A**) Schematic representation of the quantitative peptidomics analysis and (**B**–**E**) representative results. Tryptic peptides were prepared from the digestion of five selected proteins with trypsin. The resultant peptide library was aliquoted and incubated with no enzyme or different CPZ concentrations (i.e., 1, 10, and 100 nM) for 16 h at 37 °C. After incubation samples were labeled with one of four stable isotopic TMAB tags as follows: D0 for the sample without enzyme; D3 for 1 nM CPZ; D6 for 10 nM CPZ; and D9 for 100 nM CPZ. All the individual reactions were pooled and analyzed by LC-MS. Representative examples are shown for (**B**) non-substrates, (**C**) good substrates, (**D**) weak substrates, and (**E**) products of CPZ.

**Figure 6 ijms-21-08687-f006:**
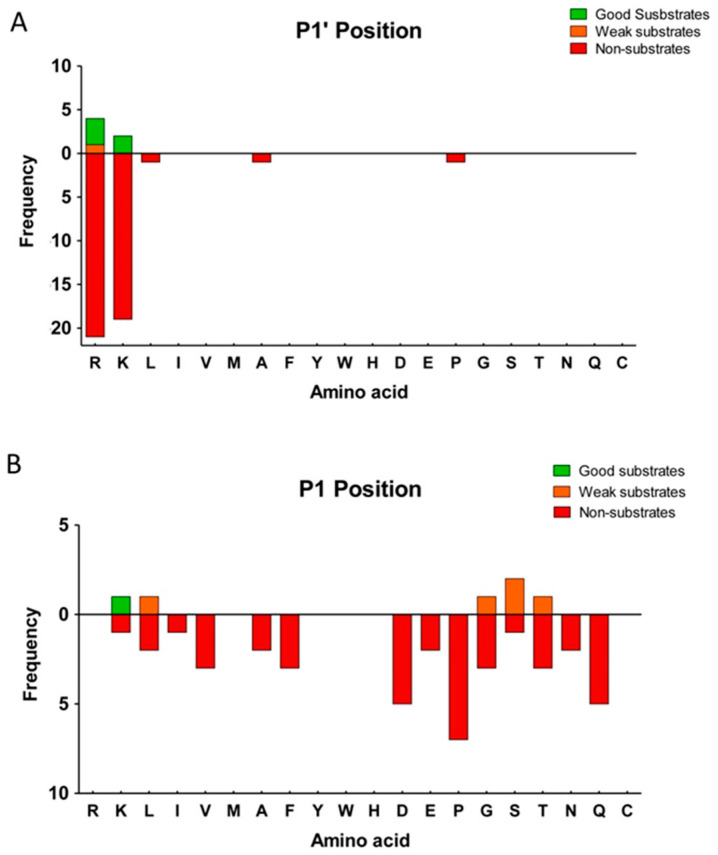
Analysis of the substrate preferences of CPZ determined using the tryptic peptide library. Substrate preferences of CPZ at (**A**) C-terminal (P1′) and (**B**) penultimate (P1) positions. The frequency for each amino acid present in P1 or P1′ is indicated for good substrates, weak substrates, and non-substrates. For P1 analysis, only substrates with permissive P1′ amino acids according to the results from (**A**) were considered.

**Figure 7 ijms-21-08687-f007:**
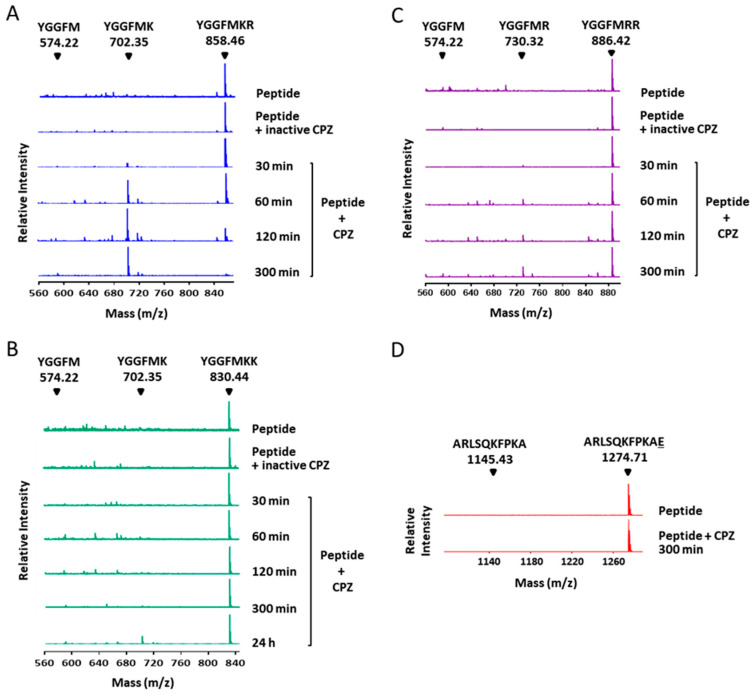
Substrate specificity of CPZ against Met-enkephalin-derived peptides. MALDI-TOF MS spectra of Met-enkephalin-derived peptides (**A**) YGGFMKR, (**B**) YGGFMKK, and (**C**) YGGFMRR treated with 100 nM CPZ at 37 °C with different incubation times. The peaks with a mass of 858.46 Da, 830.44 Da and 886.42 Da correspond to the peptides YGGFMKR (theoretical monoisotopic MH+ mass = 858.41 Da), YGGFMKK (theoretical monoisotopic MH+ mass = 830.41 Da) and YGGFMRR (theoretical monoisotopic MH+ mass = 886.42 Da) respectively. The peak with a mass of 702.35 Da generated in the presence of CPZ corresponds to the peptide YGGFMK (theoretical monoisotopic MH+ mass = 702.31 Da) produced by the cleavage of the C-terminal Arg or Lys amino acids from the peptides YGGFMKR and YGGFMKK, respectively. The peak with a mass of 730.32 Da generated in the presence of CPZ corresponds to the peptide YGGFMR (theoretical monoisotopic MH+ mass = 730.32 Da) produced by the cleavage of the C-terminal Arg from the peptide YGGFMRR. The position for the peptide YGGFM (with mass of 574.22 Da) is indicated. CPZ was inactivated by incubating the enzyme for 15 min at 90 °C to be used as a control reaction (peptide + inactive CPZ). Control samples were incubated for 24 h at 37 °C; (**D**) Spectra of the synthetic peptide ARLSQKFPKAE after 300 min of incubation in the absence (peptide) or in the presence of 100 nM CPZ (peptide + CPZ). Numbers above the major peaks indicate the monoisotopic masses of the MH+ ion (m/z) for the full-length peptide (mass = 1274.71 Da) or the peptide without its C-terminal residue, ARLSQKFPKA (mass = 1145.43 Da).

**Figure 8 ijms-21-08687-f008:**
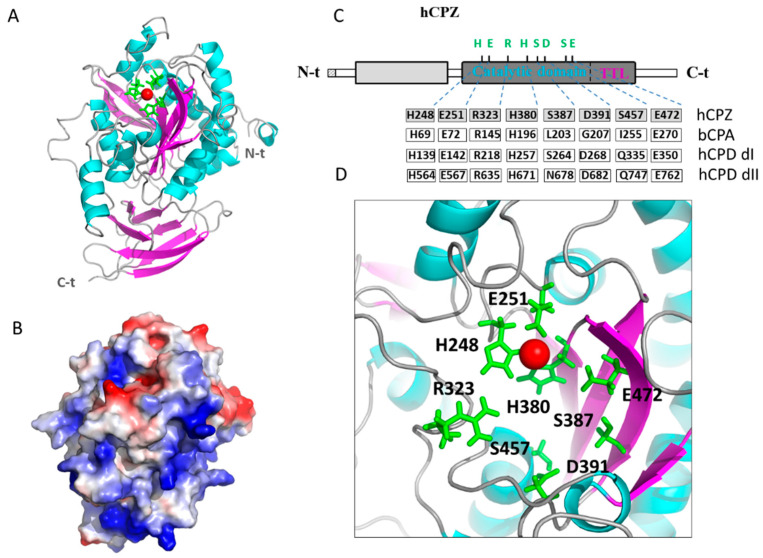
Structural modeling of the catalytic and transthyretin-like (TTL) domains of human CPZ. (**A**) Ribbon representation of human CPZ structure showing the central catalytic moiety at the top and the C-terminal TTL domain at the bottom. The side chains of the three residues involved in the Zn^2+^ binding (i.e., His248 Glu251, His380) are indicated in green; (**B**) Electrostatic surface potential distribution of the catalytic domain of human CPZ (in the same orientation as panel A). Blue indicates positive and red indicates negative charge potential; (**C**) Linear representation of the full-length human CPZ, showing the location of relevant amino acids involved in the catalytic mechanism and substrate binding (Arg323, Ser387, Asp391, Ser457, and Glu472), as well as in zinc binding (His248 Glu251, His380). Residues found in equivalent positions in bovine CPA (bCPA, as reference), in domain I of human CPD (hCPDdI) and domain II of human CPD (hCPDdII) are indicated; (**D**) Magnification of the active site of human CPZ, showing the location of these residues important for the catalytic mechanism and substrate specificity determination. The Zn^2+^ metal ion is shown in all representations as a red sphere.

**Figure 9 ijms-21-08687-f009:**
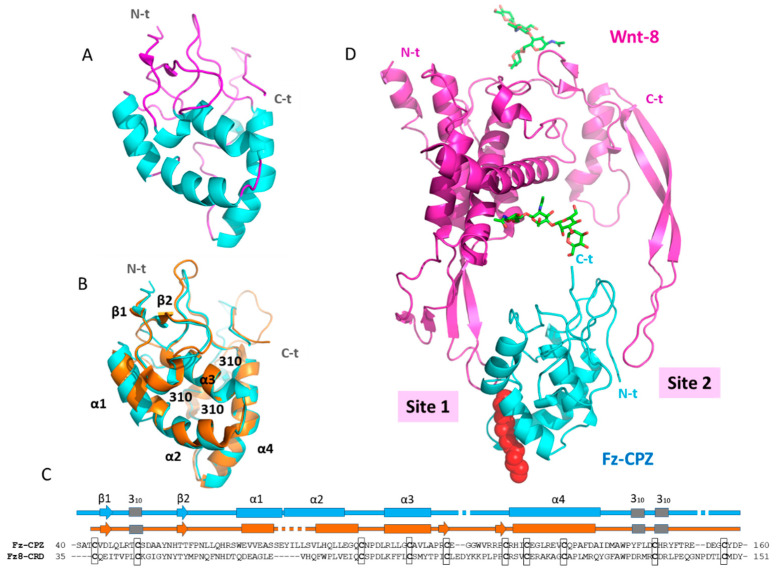
Structural modeling of the Fz domain of human CPZ and its interaction with Wnt-8. (**A**) Ribbon representation of the highly conserved N-terminal Fz domain of human CPZ; (**B**) Structural comparison of the Fz domain of human CPZ (Fz-CPZ, represented in blue) and the cysteine-rich domain of mouse Fz8 (Fz8-CRD, represented in orange); (**C**) Structure-based sequence alignment of Fz-CPZ and Fz8-CRD (represented in blue and orange respectively); (**D**) Ribbon representation of Wnt-8 in a proposed complex with the Fz domain of CPZ. The extended palmitoleic acid (PAM) group is represented with red spheres, and the most important sites of interaction between Fz-CPZ and Fz8-CRD are indicated as Site 1 and Site 2. For all proteins, the N-terminus and C-terminus are indicated as N-t and C-t, respectively.

**Figure 10 ijms-21-08687-f010:**
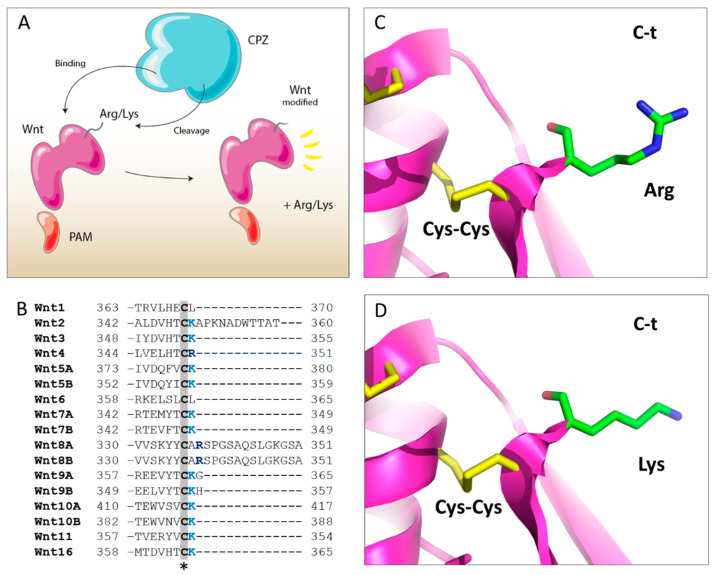
Comparison of the C-terminal region of human Wnt proteins. (**A**) Proposed model for the proteolytic cleavage of Wnt proteins by CPZ. (**B**) Sequence alignment of the C-terminal region of all human Wnt proteins. The star indicates the location of the last conserved cysteine residue, involved in a disulfide bridge formation that links the last two β-sheet regions. The C-terminal Lys and Arg residues located immediately after the last Cys residue in the majority of human Wnt proteins are shown in bright blue or dark blue, respectively; (**C,D**) Structural modeling of the C-terminal tail of XWnt-8, showing the location of the C-terminal residues identified in the majority of human Wnt proteins. Models were generated by mutating the last Ala residue found in XWnt-8 for Arg or Lys amino acids (**C** and **D**, respectively). The last Cys-Cys bond formed in XWnt-8 between Cys295 and Cys337 is indicated.

**Table 1 ijms-21-08687-t001:** Kinetic constants for the hydrolysis of dansyl-Phe-Ala-Arg by different metallocarboxypeptidases (MCPs) with B-like CP activity.

Enzyme	*K*m (µM)	*k*cat (s^−1^)	*k*cat/*K*m (µM^−1^ s^−1^)
CPZ	1905 ± 360	5.3 ± 0.6	0.0028 ± 0.0008
CPZΔFz	1667 ± 385	6.2 ± 0.8	0.0039 ± 0.0014
CPD domain I ^(a)^	319 ± 37	8.5 ± 0.5	0.027 ± 0.003
CPD domain II ^(a)^	844 ± 139	7.0 ± 0.9	0.008 ± 0.001
CPE ^(b)^	34	13	0.38

^(a)^ Data obtained from Garcia-Pardo et al. [8]; ^(b)^ Data obtained from Fricker et al. for CPE purified from bovine pituitary membranes [28].

**Table 2 ijms-21-08687-t002:** Good and weak substrates of CPZ identified using the HEK293T peptide library.

								Ratio CPZ/No Enzyme			
Type	Protein Precursor	Peptide Sequence	Z	T	Obs M	Theor M	ppm	100 nM	10 nM	1 nM	0.1 nM
Good	Histidine triad nucleotide-binding protein 1	Ac-ADEIAKAQVAR	2	1	1212.66	1212.65	14	0.02	0.94	1.08	1.00
Good	Eukaryotic translation initiation factor 5A	SAMoxTEEAAVAIKAMAK	3	3	1636.81	1636.82	−5	0.07	0.15	0.95	0.95
Good	Vimentin	AELEQLKGQGKSR	4	3	1442.78	1442.78	−3	0.18	1.00	1.03	0.95
Good	Hematological and neurological expressed 1 protein	Ac-TTTTTFKGVDPNSRNSSR	3	1	2010.00	2009.98	13	0.31	0.89	1.24	1.07
Weak	Eukaryotic translation initiation factor 5A	NMDVPNIKR	3	2	1085.56	1085.57	−6	0.45	n.d.	n.d.	0.82
Weak	Ubiquitin-60S ribosomal protein L40	IIEPSLR	2	1	826.49	826.49	0	0.60	1.00	1.13	1.04

Good substrates, peptides affected with a decrease ≥60% with the highest concentration of enzyme; weak substrates, peptides affected with a decrease ≥20% and <60% with the highest concentration of enzyme; Z, charge; T, number of isotopic tags incorporated into each peptide; Obs M, observed monoisotopic mass; Theor M, theoretical monoisotopic mass; ppm, difference between Obs M and Theor M (in parts per million); Ratio CPZ/no enzyme, the ratio in peak intensity between the sample incubated with enzyme and the sample incubated without enzyme.

**Table 3 ijms-21-08687-t003:** Products of CPZ identified using the HEK293T peptide library.

Protein Precursor	Sequence	Cleaved aa	Z	T	Obs M	Theor M	ppm	Ratio CPZ/No Enzyme			
								100 nM	10 nM	1 nM	0.1 nM
Heat shock 10 kDa protein 1	AVGSGSKGKGGEIQPVSV	K	3	3	1655.89	1655.89	2	1.30	1.16	1.28	0.93
Peptidylprolyl isomerase A	ADKVPKTAENF	R	3	3	1218.62	1218.62	−3	1.36	1.09	1.13	1.09
Nucleophosmin	EKTPKTPKGPSSVEDIKA	K	5	5	1911.00	1911.03	−18	1.46	1.00	1.23	1.15
FK506 Binding Protein	VFDVELL	K	1	1	833.45	833.45	2	1.50	1.28	ND	1.25

Products, peptides with an increase >120% with one or more concentrations of enzyme. Cleaved aa, the amino acid cleaved by CPZ to generate the observed peptide. See Table 2 for the rest of abbreviation definitions.

**Table 4 ijms-21-08687-t004:** Good substrates, weak substrates, and products of CPZ identified using the tryptic peptide library.

									Ratio CPZ/No Enzyme		
Type	Protein Precursor	Peptide Sequence	Cleaved aa	Z	T	Obs M	Theor M	ppm	100 nM	10 nM	1 nM
Good	Thyroglobulin	GQEFTITGQKR	-	2	2	1263.66	1263.60	−2	0.38	1.13	0.97
Weak	Thyroglobulin	ALEQATR	-	2	1	787.42	787.42	3	0.57	1.04	1.00
Weak	Thyroglobulin	AVKQFEESQGR	-	3	2	1277.64	1277.64	0	0.68	0.86	0.95
Weak	Bovine serum albumin	KVPQVSTPTLVEVSR	-	3	2	1638.93	1638.93	−2	0.70	0.90	0.95
Weak	Bovine serum albumin	KQTALVELLK	-	3	3	1141.69	1141.71	−22	0.73	0.80	0.89
Weak	Thyroglobulin	LPESK	-	2	2	572.31	572.32	−24	0.77	0.83	0.83
Product	Thyroglobulin	GQEFTITGQK	R	2	2	1107.56	1107.56	−4	1.52	0.78	1.04
Product	Thyroglobulin	LF	R	1	1	278.16	278.15	14	1.56	0.88	0.88

Good substrates, peptides affected with a decrease ≥60% with the highest concentration of enzyme; weak substrates, peptides affected with a decrease ≥20% and <60% with the highest concentration of enzyme; Products, peptides with an increase >120% with one or more concentrations of enzyme; Cleaved aa, the amino acid cleaved by CPZ to generate the observed peptide; Z, charge; T, number of isotopic tags incorporated into each peptide; Obs M, observed monoisotopic mass; Theor M, theoretical monoisotopic mass; ppm, difference between Obs M and Theor M (in parts per million); Ratio CPZ/no enzyme, the ratio in peak intensity between the sample incubated with enzyme and the sample incubated without enzyme.

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
