# Peer review of "Substrate Specificity and Structural Modeling of Human Carboxypeptidase Z: A Unique Protease with a Frizzled-Like Domain"

_ijms, 2020, doi:10.3390/ijms21228687_

Round 1
Reviewer 1 Report
The manuscript by Garcia-Pardo et al reports that CPZ predominantly cleaves off C-terminal arginines. It reports some preference based on the penultimate residue. It interprets these results based on an atomic model that was built on the basis of homologous enzymes.
The manuscript is well written and the conclusions are in line with the reported experimental result; I only have some minor suggestions for improvement. (i) The term 'peptidomics' is mostly used for the analysis of naturally occurring peptides. This manuscript also uses this term for the analysis of artificial peptides. I therefore suggest using the term 'peptide analysis' instead. (ii) There is no analysis of other aspects that may influence the specificity and kinetics of CPZ. If the authors did look at additional aspects (peptide length; other positions than P1 and P1'; overall charge), but did not find significant other determinants, I suggest they state this specifically. (iii) I would have appreciated an additional explanatory sentence and a reference to the quantitative peptide analysis approach using the TMAB tag in the main text. (iv) The atomic model suggests a potential binding / recognition site for the C-terminal arginine. Fig. 8 suggests it might insert into a tunnel or cavity that is lined by S387, E472, D391 and S457. Did the authors try docking an arginine moiety into this location? If so, they might consider including this as an additional confirmation of their hypothesis.
Author Response
Reviewer 1
The manuscript by Garcia-Pardo et al reports that CPZ predominantly cleaves off C-terminal arginines. It reports some preference based on the penultimate residue. It interprets these results based on an atomic model that was built on the basis of homologous enzymes. The manuscript is well written and the conclusions are in line with the reported experimental result; I only have some minor suggestions for improvement.
Answer from authors: We thank the reviewer for carefully reviewing our manuscript and for his/her positive opinion of our work.
(i) The term 'peptidomics' is mostly used for the analysis of naturally occurring peptides. This manuscript also uses this term for the analysis of artificial peptides. I therefore suggest using the term 'peptide analysis' instead.
Answer from authors: We agree that the term “peptidomics” is mostly used for the analysis of naturally occurring peptides. We used an approach developed for peptidomics to study peptides created enzymatically as well as naturally occurring peptides from HEK293 cells. Accordingly, we have replaced the term “peptidomics” with “peptidomics analysis” or with “peptidomics approaches” in the revised text.
(ii) There is no analysis of other aspects that may influence the specificity and kinetics of CPZ. If the authors did look at additional aspects (peptide length; other positions than P1 and P1'; overall charge), but did not find significant other determinants, I suggest they state this specifically.
Answer from authors: We thank the reviewer’s suggestion. Previous studies have shown that the P1′ position is the major specificity determinant for M14 MCPs, and that amino acids at the P1 position also may have an effect on the substrate specificity [1-4]. For this reason and due to the limited number of substrates identified, we did not consider other positions than P1´ or P1 in the present study. In a similar manner and based on these previous studies, we do not expect an impact of the chain length or overall charge of the peptide.
- Garcia-Pardo, J.; Tanco, S.; Diaz, L.; Dasgupta, S.; Fernandez-Recio, J.; Lorenzo, J.; Aviles, F.X.; Fricker, L.D. Substrate specificity of human metallocarboxypeptidase d: Comparison of the two active carboxypeptidase domains. PLoS One 2017, 12, e0187778.
- Tanco, S.; Zhang, X.; Morano, C.; Aviles, F.X.; Lorenzo, J.; Fricker, L.D. Characterization of the substrate specificity of human carboxypeptidase a4 and implications for a role in extracellular peptide processing. J Biol Chem 2010, 285, 18385-18396.
- Lyons, P.J.; Fricker, L.D. Substrate specificity of human carboxypeptidase a6. J Biol Chem 2010, 285, 38234-38242.
- Sebastian Tanco, Julia Lorenzo, Javier Garcia-Pardo, Sven Degroeve, Lennart Martens, Francesc Xavier Aviles, Kris Gevaert, Petra Van Damme. Mol Cell Proteomics 2013, Aug;12(8):2096-110.
(iii) I would have appreciated an additional explanatory sentence and a reference to the quantitative peptide analysis approach using the TMAB tag in the main text.
Answer from authors: According to the Reviewer suggestion, a sentence explaining the quantitative peptide analysis approach using TMAB tags and additional references have been included in the main text of the manuscript. According to such modifications, the numbering of the references has been changed.
(iv) The atomic model suggests a potential binding / recognition site for the C-terminal arginine. Fig. 8 suggests it might insert into a tunnel or cavity that is lined by S387, E472, D391 and S457. Did the authors try docking an arginine moiety into this location? If so, they might consider including this as an additional confirmation of their hypothesis.
Answer from authors: We appreciate the reviewer’s suggestion. However, we did not perform the docking of the arginine into the catalytic domain of CPZ. In the revised manuscript, we have expanded the discussion with additional information and a reference to our previous work on CPD, in which an extensive docking experiment has been performed. Please see lines 434-436.
Reviewer 2 Report
The authors have performed a detailed characterization of the human CPZ by biochemical experiments and modeling studies. I have a couple of suggestions for minor edits
- Please change "similitude" to "similarity". The manuscript will benefit from being edited by a native English speaker.
- The suggested model of the CPZ cleaving the C-terminal Arg/Lys from the Wnt protein is not clear from the current figures. A schematic representation will be helpful.
- The Histidine and the Glu residues are active site residues. Please change the term "three protein ligands" in line 332-333. This is confusing.
Author Response
Reviewer 2
The authors have performed a detailed characterization of the human CPZ by biochemical experiments and modeling studies. I have a couple of suggestions for minor edits
We would like to thank to the reviewer for the valuable comments and suggestions.
Please change "similitude" to "similarity". The manuscript will benefit from being edited by a native English speaker.
Answer from authors: We thank the reviewer for pointing out the error. The word “similitude” has been replaced by “similarity” in the revised version of the manuscript. A native English speaker has carefully edited the entire manuscript and made a number of additional changes.
The suggested model of the CPZ cleaving the C-terminal Arg/Lys from the Wnt protein is not clear from the current figures. A schematic representation will be helpful.
Answer from authors: We appreciate the reviewer’s suggestion. We have included a new scheme in Figure 10 (see panel a) in the revised version of the manuscript.
The Histidine and the Glu residues are active site residues. Please change the term "three protein ligands" in line 332-333. This is confusing.
Answer from authors: We considered the reviewer’s suggestion, but technically these residues are not considered active site residues involved in catalysis. Therefore we have changed the term “three protein ligands” to “residues involved in the binding” of the catalytic zinc ion (lines 333-334 in the revised manuscript).